# Simulation Study of the Impact of COVID-19 Policies on the Efficiency of a Smart Clinic MRI Service

**DOI:** 10.3390/healthcare10040619

**Published:** 2022-03-25

**Authors:** Francesca Sala, Mariangela Quarto, Gianluca D’Urso

**Affiliations:** Department of Management, Information and Production Engineering, University of Bergamo, Via Pasubio 7/b, 24044 Dalmine, Italy; francesca.sala@unibg.it (F.S.); gianluca.d-urso@unibg.it (G.D.)

**Keywords:** Discrete-Event Simulation, FlexSim Healthcare, outpatient clinic, Gruppo San Donato, COVID-19, anti-contagion policies

## Abstract

The present study examines the impact of the policies against the proliferation of SARS-CoV-2 on outpatient facilities through a direct comparison of the key performance indicators measured in an ordinary and pandemic scenario. The subject of the analysis is a diagnostic imaging department of a Smart Clinic (SC) of Gruppo San Donato (GSD). The operations are virtually replicated through a Discrete-Event Simulation (DES) software called FlexSim Healthcare. Operational and productivity indicators are defined and quantified. As hypothesized, anti-contagious practices affect the normal execution of medical activities and their performance, resulting in an unpleasant scenario compared to the baseline one. A reduction in the number of diagnoses by 19% and a decrease in the utilization rate of the diagnostic machine by 21% are shown. Consequently, the development of strategies that restore balance and improve the execution of outpatient activities in a pandemic setting is necessary.

## 1. Introduction

The current SARS-CoV-2 pandemic has had substantial consequences for businesses, particularly in the medical field. As the pandemic evolved, hospitals became saturated, causing overcrowding of emergency departments (EDs) and intensive care units (ICUs). Since March 2020, ordinary elective activities have been interrupted or postponed, depending on the country’s level of emergency, denoting a one-third drop in regular health services delivery [1]. Accordingly, governments introduced public health measures to mitigate the contagions and alleviate the massive pressure on medical organizations, reshaping physical assets as well as social and relational dynamics. The adhesion to such policies ensures a fresh start in the non-urgent care supply process, inevitably impacting service performance: operational and productivity metrics and, as a result, financial metrics are expected to be adversely influenced [2]. The development of strategies that restore balance and improve the execution of outpatient activities in pandemic settings is thus crucial.

Several studies have focused on the influence of COVID-19 on EDs and ICUs [3,4,5,6,7,8,9], while less emphasis has been given to outpatient services [2,10,11]. This is a consequence of the non-urgent services being locked down for extended periods, while EDs and ICUs have faced ongoing emergencies. For ensuring a safe restart of the entire healthcare system, the reopening of outpatient clinics respecting the COVID-19 anti-contagion guidelines allows a reduction of the pressure on hospitals. In the face of what has been stated, the present study aims at exploring the potential effects of the policies against the proliferation of SARS-CoV-2 in outpatient medical facilities. Specifically, this research investigates how the guidelines, introduced as a consequence of the pandemic emergency, modify patient flow and the key performance areas of a new Smart Clinic (SC) of Gruppo San Donato (GSD).

The method pursued to perform this study is Discrete-Event Simulation (DES) [12,13]. The technique is based on the concept of modelling the operations of a real-world system as a discrete series of events over time. DES allows us to gain knowledge of the process functions without committing resources for their implementation or physically interacting with them. Moreover, the approach represents a valuable tool for evaluation and comparison, quantifying how well a system behaves compared against specific criteria or even other systems, enabling the user to interpret input conditions, process them and estimate their effects. The DES method, in the last 30 years, has represented one of the most valuable methodologies for the analysis of the healthcare process. A DES model used for increasing throughput in an emergency department (ED) in the USA was developed [14,15,16]. The development of a simulation model is very useful in healthcare systems, but it is impossible and impractical to have a DES model for an entire hospital, as models are intended to be simplifications [17]. An appropriate level of abstraction and a precise scope must be chosen [18]. Given its great flexibility, DES has proven to be a worthy instrument for the analysis of care processes [19,20,21,22,23], as it allows the management of stochastic activities comprehensively [24]. The selected software package is FlexSim Healthcare, which has been successfully used in the resolution of a variety of medical issues [25,26,27,28,29]. The adaptation of the FlexSim simulation to healthcare environments has been undertaken, but the use of this software has not yet been explored to comprehend the current emergency.

In this study, a virtual replica of a real-world diagnostic process is proposed, comparing two different scenarios (ordinary and pandemic) in terms of the most significant operational and productivity indicators. The analysis allows us to observe the impact of a pandemic scenario on the performance of an outpatient clinic, underlining possible improvements.

## 2. Characteristics of a Smart Clinic

The object of this study is an Italian private healthcare facility, a newly opened (not operative at the time of analysis) Smart Clinic (SC) of Gruppo San Donato (GSD). This is a revolutionary outpatient polyclinic that goes beyond the borders of the hospital and moves as close to people as possible. Indeed, it is located within a shopping mall. Among the medical specialties that the facility plans to offer, the magnetic resonance imaging (MRI) diagnostic discipline is the topic of our study.

### Process Conceptualization

The dynamics of patient flow inside the MRI diagnostic unit of the SC are conceptualized through the Business Process Model and Notation (BPMN 2.0) tool. Figure 1 reports the activities and stakeholders involved in the MRI diagnostic examination process. In addition to the patient, a receptionist, a registered nurse, a physician and an MRI technician are involved. Figure 2 describes the patient flow, clustered into five distinct sub-processes, namely registration, anamnesis, examination, medical results and leaving the clinic. 

The process begins on the day of the diagnostic inspection. The patient enters the medical facility and reaches the acceptance desk. There, the individual is enrolled by the receptionist and pays for the service. At the end of the registration, the patient is escorted by the nurse towards the doctor’s studio, where the preliminary consultation takes place; the physician gathers information about the patient’s medical history and collects the written informed consent (given by the receptionist on the day of the appointment booking). Afterwards, the patient goes to the dressing room and then to the examination room, where he or she finally undergoes the MRI investigation with the technician. Occasionally, the diagnostic inspection is preceded by the contrast medium injection (CMI). While the patient returns to the dressing room, the doctor and the technician exchange opinions on the test just performed. Then, the physician interprets the MRI images and communicates the diagnosis to the patient, who moves to the exit. The medical staff is ready to handle a new diagnostic case.

## 3. Methodology

The conceptualized flowchart of the GSD SC is translated into a computer model, designed through the simulation software FlexSim Healthcare v. 20.1. Firstly, the 3D model interface is developed respecting the space boundaries and distances. Secondly, throughout a process flow interface, the logic and interactions are applied to the 3D model. Therefore, the output of the designing activity is a discrete, stochastic and dynamic model simulating the ordinary flows inside an MRI diagnostic ward.

### 3.1. Baseline Scenario

The 3D items necessary to guarantee the execution of the diagnostic imaging service are an entry/exit door, a registration desk with a telephone, several seats (located in the two waiting rooms), two workstations (one for the doctor and one for the technician), some dressing rooms, a triage area (dedicated to the contrast medium injection (CMI)) and an MRI machine. Some task executers, denoting the four workers, are included in the 3D model. The patient activities and the interactions among the objects are implemented in separate process flows. Figure 3 shows the GSD SC baseline 3D model.

### 3.2. Pandemic Scenario

The SC system is reproduced in compliance with protocols minimizing the risk of contagion from SARS-CoV-2. The baseline model is modified in accordance with the existing guidelines about the management of a post-emergency COVID-19 medical clinic, in particular the regulations issued by *Federazione Italiana Medici di Famiglia* [30] and the measures drawn up by GSD [31]. 

The documents enforce the adoption of collective and individual protection measures. Dividers and panels are installed to maintain a physical separation between the patient and the specialists; sanitizer, face masks and surgical gloves are introduced, generating only a marginal effect on the patient’s flow. Organizational risk prevention actions are implemented. Specifically, these activities are categorized as operations assuring a safe entrance into the clinic (hand sanitation, temperature inspection) and operations assuring a safe exam (equipment sterilization and room disinfection) (Figure 4). Novel entrance and exit pathways are planned.

### 3.3. Model Data

The virtual models are fed with the information provided by the GSD personnel during a preliminary meeting. The missing evidence is hypothesized based on literature findings [32] and validated by an expert in the radiological field. The mentioned referential canals supply stochastic data, allowing the replica to capture the irregularity and variability of the healthcare process and delivering outcomes more consistent with reality. The adequacy of the models with the intended application is assessed in terms of subjective reviews (face validity). Face validity is the extent to which a test is subjectively viewed as covering the concept it purports to measure. It refers to the transparency or relevance of a test as it appears to test participants; a test can be said to have face validity if it “looks like” it is going to measure what it is supposed to measure. Face validity is achieved when a model represents reality at least on the surface or in appearance [33,34], and it is evaluated by an individual with a deep knowledge of the analysed system. The main characteristics of the GSD stakeholders are reported in Table 1. 

Supplementary information about the distinct diagnostic flows undertaken by the patient is provided. The people entering the SC to perform an MRI investigation are distributed into two major specialties: single and multiple body region analysis. The first category is faster and more frequent (+80%) than the second one. A supplementary partition is due to the possible CMI. If in the hospital the probability of performing a CMI is about 50% of cases, in the clinic this is reduced to 30%. The statistical distributions describing the time parameters of the diagnostic process are detailed in Table 2 and Table 3. According to the data provided by the GSD experts, the most used distribution is the normal one (described by a mean value and a standard deviation). The MRI exam information better fits a uniform distribution (described by a minimum and a maximum value).

### 3.4. Key Performance Indicators

Key performance indicators (KPIs) allow the supervision of several metrics during the simulation of the process. The selected KPIs are categorized into operational and productivity indicators (Table 4). The operational KPIs are related to the activities involving the patient, and the productivity KPIs evaluate the staff and the equipment utilization in the process.

The productivity indicator denoting the MRI device inactivity is classified into further indicators according to the origin of MRI idleness. The first downtime is related to the mismatch between the opening time to the public (11 h) and the MRI operating time (12 h). The second and third downtimes refer to the first and last patient entering the outpatient clinic. Every working day, before performing the actual exam, the first patient conducts some tasks (registration and anamnesis) that postpone the machine deployment; in addition, at the end of every working day, there is always a time margin between the moment in which the last person leaves the MRI bed and the moment at which the clinic closes to the public. CMI downtime includes the examination delays caused by CMI activity, while the staff consultation downtime includes the device interruptions due to the meeting between physician and technician after every diagnostic investigation. Sanitation downtime occurs after every diagnostic examination; the MRI machine remains in an idle state due to the disinfection activity. Other downtime comprises all the idle time not attributable to the previous categories, such as MRI stops due to delays in the arrival of patients and the temporary inability to escort people to the examination room.

### 3.5. Improvement of the Pandemic Model 

In a pandemic environment, the SC system is expected to behave adversely in terms of operational and productivity metrics, since the adopted anti-contagion measures deny the SC the possibility to operate at its maximum capacity. In compliance with current anti-contagion regulations, three simultaneous changes are proposed and virtually tested. Firstly, the flow of individuals who merely have to schedule a medical visit is replaced by telephone or online reservations, reducing the influx of people and the risk of creating crowds. Secondly, the execution of the medical reporting activity on site is discarded. The diagnosis communication might be accomplished in a variety of other forms (electronic health record compilation and telematic appointments), reducing the patient’s stay in the clinic. The current change shortens the waiting time of the patient, promoting the possibility to treat more subjects per day while decreasing the contamination risk. Lastly, the nurse is relieved from the anamnesis activity and partially absolved from the task of escorting the individual from the reception to the diagnostic department, since the patient may not always be able to complete the visit independently. Thus, higher priority activities can be accomplished.

## 4. Results and Discussion

### 4.1. Results

The models are simulated over a period of seven working days. The radiological unit operates for twelve hours (from 8.30 am to 8.30 pm), with public accessibility of eleven hours (from 9.00 am to 8.00 pm). The results of the analysis of the three simulative scenarios are summarized in the following figures (Figure 5, Figure 6, Figure 7 and Figure 8). Specifically, Figure 5 shows the overall stay-time of the patients as a function of the different activities; the pandemic scenarios reduce the amount of time spent by the patients inside the clinic. The receiving-direct-care time remains the activity in which the patients are involved for the longest time. At the same time, the COVID-19 scenario generates a natural reduction of the treated patients due to the restrictions, and the suggested improvements allow the reduction of the loss of patient volume (Figure 6). Consequently, the same trend can be noted in MRI utilization (Figure 7). From the staff utilization point of view, the technician is the busiest; the pandemic scenario generates a reduction in the amount of time that the staff dedicates to the main activities, since two new members are added, and during their activities (in particular the cleaner activities), the medical staff cannot continue the examinations. In the pandemic scenario, the level of utilization of the staff is affected both by the lower number of processed patients and by the suggested improvements, reorganizing the medical activities (Figure 8).

### 4.2. Discussion

The operational indicators show that, on average, in the baseline scenario, more than one third of the time spent by the patients in the clinic is unproductive. The pandemic scenario behaves comparably and, unexpectedly, the facility stay-time is characterized by a slight reduction, entirely attributable to a decrease in the idle time as an indirect consequence of the adoption of anti-SARS-CoV-2 regulations. In fact, the pandemic system is characterized by a lower patient volume, partially due to the people with a body temperature equal to or higher than 37.5 °C, who are not allowed to continue their diagnostic pathway. Given the absence of a person, the next individual joining the process performs their operations in less time, thus reducing the overall queues of the system. Moreover, medical resources are more available and tend to shrink patient waiting periods. It is evident that sanitization activities do not lengthen the waiting time; on the contrary, the preventive and protective measures appear to decrease patient inoperativeness. As assumed in the second enhancement proposal, the facility time in the improved COVID-19 scenario is shortened even more compared to the previous two contexts. The result achieved is largely attributable to the removal of the on-site diagnosis communication.

Focusing on productivity indicators, the shift from the baseline setting to the pandemic one results in a broad decrease in the volume of treated patients. The changes implemented to improve the COVID-19 scenario allow an increment of 8% in capacity.

Regarding the diagnostic machine, an overall utilization rate drop is highlighted. In the ordinary scenario, the MRI usage amounts to 9.12 h daily (76%). The main causes of inefficiencies are last patient idle and other idle, which together generate 12% of machine downtime. All the other causes of MRI inoperativeness are related to system boundaries (e.g., public closing) or due to unavoidable activities (e.g., CMI). In the pandemic framework, the MRI utilization rate lowers to 60% per day. The downtime due to public closure remains unchanged, and barring changes in the maximum working hours for the specific facility, this is constant in each scenario. The MRI examination of the first patient is delayed by one minute due to the introduction of sanitation checks at the entrance. Conversely, the last patient downtime extends, highlighting the inability of the system to exploit that time to process a further patient. The CMI downtime diminishes due to the decreased volume of CMI cases and the overall reduction of treated patients. In the presence of the pandemic, staff consultation downtime is absorbed by MRI sanitation downtime, since both idle times occur simultaneously, though sterilisation lasts longer than staff consultation. Other downtimes are characterized by a dramatic rise; the idleness due to the delays in the arrival of the patients and the unavailability of the staff becomes marginal, although occasional inactivity due to the patients leaving the system because of body temperature is preponderant. The COVID-19 scenario exhibits a deterioration in MRI performance, and the fraction of justifiable downtime is largely narrowed. Together, the two main causes of inefficiency constitute 41% of the device inactivity, while sanitization activity represents 23%. Regarding the improved COVID-19 setting, the MRI utilization rate rises to 66% per day. The public closing and first patient clusters remain unchanged. The equipment idleness related to the CMI and MRI sanitation increases, since their values grow with the productivity of the diagnostic device apparatus. On the contrary, last patient idle and other idle reduce. Therefore, the three proposed improvements appear to slightly boost MRI productivity, reducing the inoperativeness of the diagnostic machine. The source of actual inefficiency is 26% device idleness, and the sanitization rises to 28%.

The results of the productivity indicators show a general and noticeable underuse of the medical staff. In each scenario, the technician is always the most highly engaged resource; in fact, he or she is directly related to the utilization of the MRI. The other workers are minimally employed. The underuse of shared resources (receptionist, clerk and cleaning staff) is consistent with the GSD’s choice of directing the resources to other tasks within the SC.

The three proposed measures (Section 3.5) are not solely based on the system’s ability to achieve the best performance but also on the evaluation of certain applicability criteria, such as cost and ease of implementation. Indeed, the emphasis is placed on low economical effort; in general, the improvements should be implemented easily from both the medical staff and patient perspectives. The activities of the healthcare operators are reorganized, while keeping patient process flow nearly unchanged. Furthermore, the improvements of the COVID-19 model should not affect the quality of provided care. However, the second change is responsible for altering the patient’s experience and engagement in the process; establishing an appropriate procedure for performing tasks in a telematic way ensures that a certain level of quality is maintained.

## 5. Conclusions and Further Developments

The present article explores the possible consequences of the policies against the proliferation of SARS-CoV-2 in outpatient medical facilities. The patient flow in a diagnostic imaging department of a GSD SC is virtually simulated through FlexSim Healthcare v. 20.1 software.

The baseline scenario represents the healthcare activities in a daily routine. The diagnostic department achieves good results in terms of operational and productivity KPIs. In particular, the MRI machine employment is equivalent to 76%. Downtimes are discerned and justified. Then, the behaviour of the GSD SC is implemented for simulating the contemporary pandemic environment, meaning that the anti-contagion regulations enrich the original simulation. Highlighting potential abnormalities due to the presence of the COVID-19 factor is immediately feasible; the operational KPIs seem to be positively affected, while the productivity KPIs are negatively influenced. On one side, there is a global reduction in patient stay-time and idle time. On the other side, reductions in the number of people treated by the diagnostic department, MRI machine usage and staff utilization are assessed. In addition, the actual inefficiencies prompting the device downtime become more influential. 

The implementation of additional adequate measures proves that the adverse effects of the pandemic might be somehow contained. The proposed improvements do not allow achievement of the level of performance simulated in the absence of the pandemic; however, they make the system more competitive. In fact, compared to the COVID-19 setting, an increment of the number of diagnoses of 8% and a growth in the MRI utilization of 10% are assessed. These last results display the need to refine the current COVID-19 policies, those defined at both a local and national level, to enhance the efficiency of the outpatient clinic. The reorganization of the medical activities and the proposal of further advancements must proceed. Once again, the new alternatives should be evaluated with specific criteria to assess their actual applicability in medical practice. 

To conclude, the developed simulation model gives a general overview of the MRI diagnostic examination process, enabling the collection of data useful for the KPI evaluation. Further studies should extend the analysis on the adequacy of the models with the intended applications from qualitative means to quantitative means. With the opening of the GSD SC, it will be possible to conduct a data validation, bypassing the main limitation of the model. Furthermore, the presence of other healthcare services within the facility might affect the described process. A further model implemented by logistics and interactions related to the other departments and processes will allow obtaining a more representative model.

## Figures and Tables

**Figure 1 healthcare-10-00619-f001:**
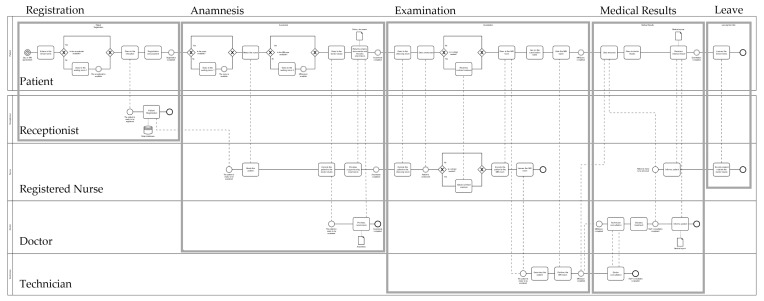
MRI examination process.

**Figure 2 healthcare-10-00619-f002:**
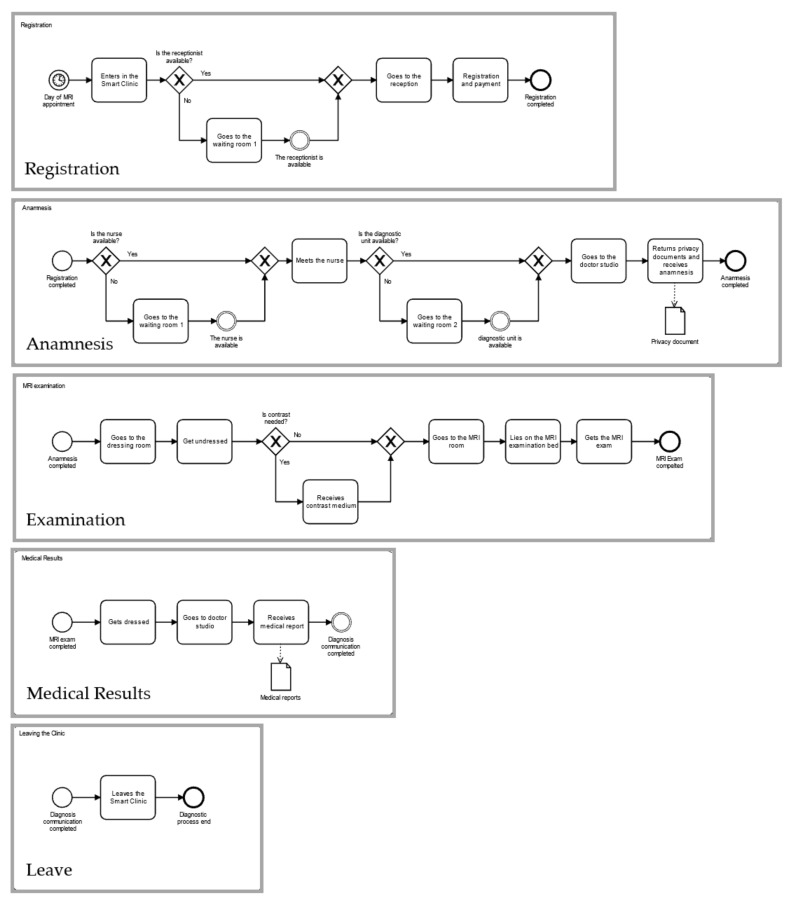
Patient macro-activities of the MRI examination process.

**Figure 3 healthcare-10-00619-f003:**
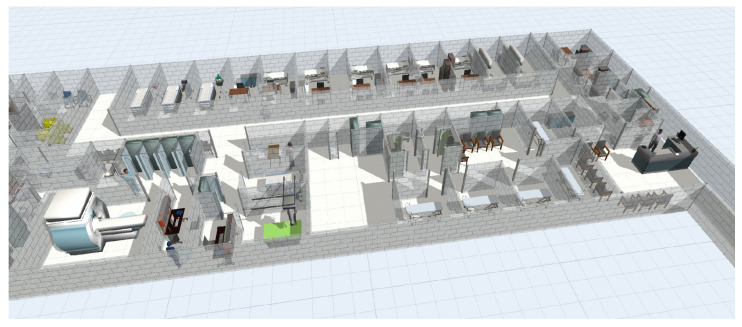
3D Baseline model of the GSD SC.

**Figure 4 healthcare-10-00619-f004:**
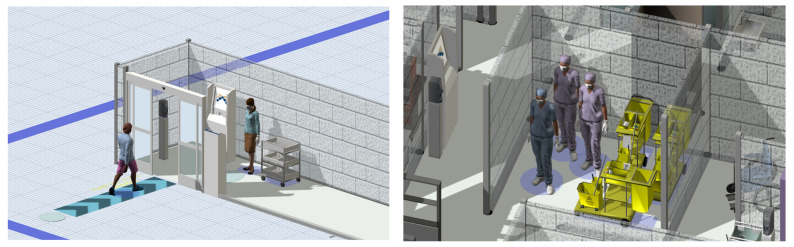
3D views of the pandemic model of the GSD SC. On the left, patient and clerk performing the pre-registration step (hand disinfection, temperature control and protective equipment wearing) at the SC entrance. On the right, cleaning staff and carts dedicated to the sanitation activities.

**Figure 5 healthcare-10-00619-f005:**
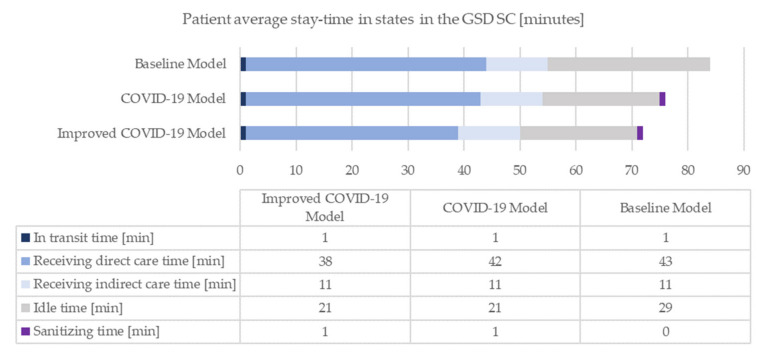
Patient average stay-time in the GSD SC.

**Figure 6 healthcare-10-00619-f006:**
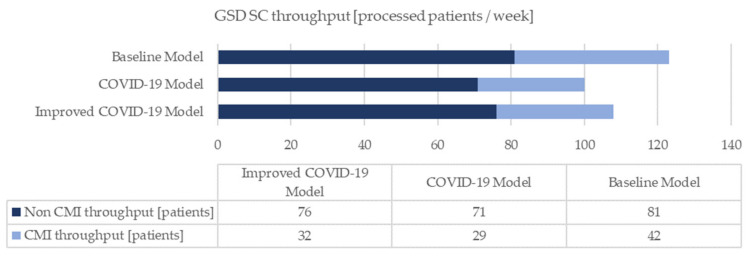
GSD SC throughput.

**Figure 7 healthcare-10-00619-f007:**
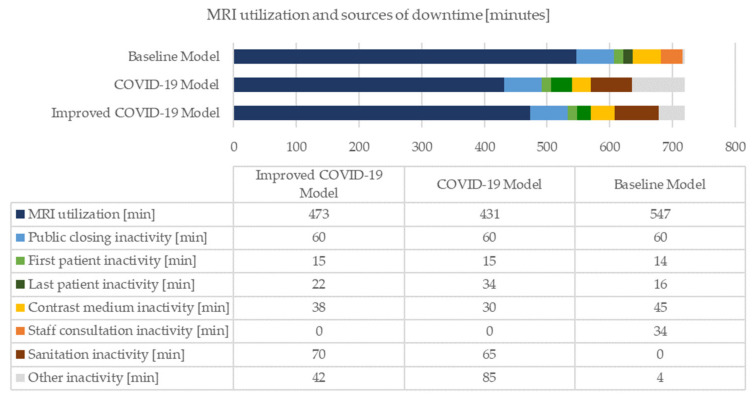
MRI utilization and sources of downtime.

**Figure 8 healthcare-10-00619-f008:**
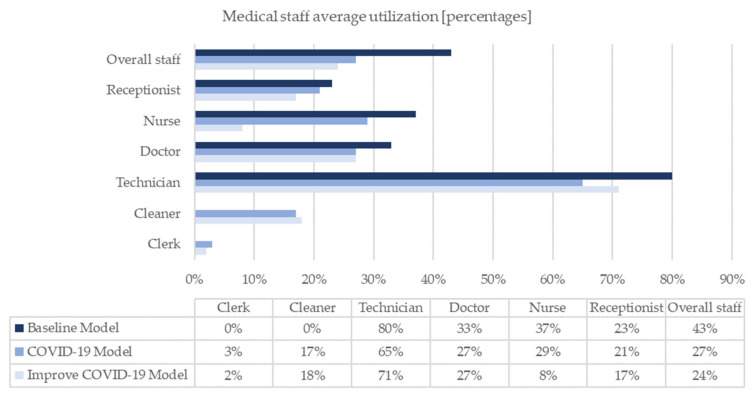
Medical staff average utilization.

**Table 1 healthcare-10-00619-t001:** Volumes, calendar and states of the process participants.

Participant	Baseline Model Units	COVID-19 Model Units	Time Schedule	States
Patient	123 *	100 *	9.00 a.m.–8.00 p.m.	Receiving Direct Care
				Receiving Indirect Care
Receptionist	1	1	8.30 a.m.–8.30 p.m.	Registering Patient
				Booking Appointments
				Clinic Closing/Opening
Registered nurse	1	1	9.00 a.m.–8.00 p.m.	Anamnesis
				CMI
				Patient Information
Physician	1	1	9.00 a.m.–8.00 p.m.	Anamnesis
				Staff Consultation
				Patient Information
Technician	1	1	8.30 a.m.–8.30 p.m.	Examination
				Staff Consultation
				Clinic Closing/Opening
COVID-19 clerk	0	1	9.00 a.m.–8.00 p.m.	Sanitization
Cleaning staff	0	1	9.00 a.m.–8.30 p.m.	Sanitization

* The number of patients treated in the SC is not an input variable. The volume of patients is estimated as an output, based on the characteristics and performances of the developed model. Only one individual input factor is selected: the arrival frequency. The virtual generation of people in the process is equally spaced over time. The derived hypothesis is acceptable, since the instance replicates a predictable service (an election examination).

**Table 2 healthcare-10-00619-t002:** Time distribution of the key process activities—Baseline Model.

Process Activity	Distribution (Parameters) (s)	Participant Involved	State of the Participant
Appointment booking	Normal (300,13)	Patient	Receiving Indirect Care
		Receptionist	Booking Appointments
Registration and payment	Normal (300,13)	Patient	Receiving Indirect Care
		Receptionist	Registering Patient
Anamnesis	Normal (300,13)	Patient	Receiving Direct Care
		Nurse	Anamnesis
		Doctor	Anamnesis
Undress CMI *	Normal (180,11)	patient	Receiving Indirect Care
	Uniform (300,600)	Patient	Receiving Direct Care
		Nurse	Contrast Medium Injection
Patient examination	Normal (120,20)	Patient	Receiving Direct Care
		Technician	Examination
MRI exam (single area) *	Uniform (900,1800)	Patient	Receiving Direct Care
		Technician	Examination
MRI exam (multiple areas) *	Uniform (2700,3600)	Patient	Receiving Direct Care
		Technician	Examination
Dress	Normal (180,11)	Patient	Receiving Indirect Care
Check MRI correctness	Normal (120,20)	Technician	Staff Consultation
		Doctor	Staff Consultation
Diagnosis	Normal (300,13)	Patient	Receiving Direct Care
		Doctor	Patient Information
		Nurse	Patient Information

* Data hypothesized based on literature research.

**Table 3 healthcare-10-00619-t003:** Time distribution of the key additional process activities—COVID-19 Model.

Process Activity	Distribution (Parameters) (s)	Participant Involved	State of the Participant
Hand sanitizing	Uniform (5,10)	Patient	Sanitization
Temperature scanning	Uniform (8,12)	Patient	Sanitization
		Clerk	Sanitization
Wear mask and gloves	Normal (300,13)	Patient	Sanitization
CMI site cleaning	Normal (60,2)	Cleaner	Sanitization
MRI cleaning	Normal (300,10)	Cleaner	Sanitization
Dressing room cleaning	Normal (120,5)	Cleaner	Sanitization

**Table 4 healthcare-10-00619-t004:** KPIs of the GSD SC and their definition.

KPIs		Definition
OPERATIONAL INDICATORS	Facility time (min)	Time spent by patients inside the facility
	Receiving Direct Care time (min)	Time spent by patients performing tasks that add value to the diagnostic process
	Receiving Indirect Care time (min)	Time spent by patients performing indispensable tasks without added value
	In transit time (min)	Walking time of patients
	Idle time (min)	Time spent by patients in idle or non-value-added tasks
	Sanitizing time (min)	Time spent by patients in sanitation activities
PRODUCTIVITY INDICATORS	Throughput (patients)	Number of treated patients/week
	CMI throughput	Number of CMI patients/week
	MRI utilization (min (%))	Time of MRI use
	MRI downtime (min (%))	Time of MRI downtime
	Public closure (min (%))	Downtime owing to the mismatch between availability and activation of MRI machine
	First patient (min (%))	Downtime owing to the daily first patient
	Last patient (min (%))	Downtime owing to the daily last patient
	CMI (min (%))	Downtime owing to the CMI
	Staff consultation (min (%))	Downtime owing to the technician–doctor consultation
	Sanitation (min (%))	Downtime owing to the MRI sterilisation
	Other (min (%))	Other downtimes not attributable to the previous groups
	Staff utilization (%)	Time of staff use in working activities
	Receptionist (%)	Time of receptionist use in working activities
	Registered Nurse (%)	Time of registered nurse use in working activities
	Physician (%)	Time of physician use in working activities
	Technician (%)	Time of technician use in working activities
	Cleaning Staff (%)	Time of cleaning staff use in working activities
	Clerk (%)	Time of clerk in working activities

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
