# Peer review of "Simulation Study of the Impact of COVID-19 Policies on the Efficiency of a Smart Clinic MRI Service"

_healthcare, 2022, doi:10.3390/healthcare10040619_

Round 1
Reviewer 1 Report
The authors present the research results on the possible consequences of the policies against the proliferation of SARS-CoV-2 in ambulatory medical facilities.
Different articles analyzed in recent years show that Discrete Event Simulation (DES) is mainly applied to improve the healthcare supply chain and mainly for the emergency department. But given the situation presented by the SARS-CoV-2 pandemic, the various logistics activities must be improved, such as instrument sterilization. As of the research carried out, these results can compare it with the different simulation techniques applied in the health care supply chain (system dynamics (SD), discrete event simulation (DES), agent-based modeling) and introduce important activities as sterilization.
Author Response
Reviewer 1
C: The authors present the research results on the possible consequences of the policies against the proliferation of SARS-CoV-2 in ambulatory medical facilities.
Different articles analyzed in recent years show that Discrete Event Simulation (DES) is mainly applied to improve the healthcare supply chain and mainly for the emergency department. But given the situation presented by the SARS-CoV-2 pandemic, the various logistics activities must be improved, such as instrument sterilization. As of the research carried out, these results can compare it with the different simulation techniques applied in the health care supply chain (system dynamics (SD), discrete event simulation (DES), agent-based modeling) and introduce important activities as sterilization.
The instrument sterilization was already considered in the manuscript. The explanation about the additional organizational and risk prevention operation (particularly, instrument sterilization) was not described in a clear way. Section 3.2 has been improved, identifying the sterilization activity as follows:
“The documents enforce the adoption of collective and individual protection measures. Dividers and panels are installed to maintain a physical separation between the patient and the specialists; sanitizer, face masks and surgical gloves are introduced generating only a marginal effect on the patient’s flow. Organizational risk prevention actions are implemented. Specifically, these activities are categorized as operations assuring a safe entrance into the clinic (hand sanitation, temperature inspection) and operations assuring a safe exam (equipment sterilization and room disinfection) (Figure 4). Novel entrance and exit pathways are planned.”
Regarding the selection of the simulation tool, discrete-event simulation was chosen due to its successful implementation in the healthcare setting observable in literature. This manuscript aims to develop a virtual replica of a diagnostic process for evaluating the differences related to the influence of the pandemic in terms of performances. Given the specificity of this objective, the introduction of a comparative analysis between different simulation methods represents a good starting point for further research; in fact, the development of a such work requires a completely dedicated manuscript and not just a paragraph section.
Reviewer 2 Report
A very interesting research idea, congratulations to the authors.
I miss the literature justification for undertaking the topic of defining the research gap, apart from the obvious Covid that fits everything.
There are colors in picture 1, no explanation anywhere.
There is no literature analysis of similar studies.
Correctly designed model, is there a clinic to compare it with?
The combination of results and discussion is a bad idea. It is worth discussing the results and then discussing the discussion.
No reference to the limitations of the study noted by the authors.
Distressingly few references to literature.
Author Response
Reviewer 2
C: Very interesting research idea, congratulations to the authors. I miss the literature justification for undertaking the topic of defining the research gap, apart from the obvious Covid that fits everything.
The literature justification for undertaking the topic was improved and is reported from line 40 to line 47, including the following references:
- Bovim et al, 2022
- Garcia-Vicuna et al, 2020
- Le Lay et al, 2020
- Melman et al, 2021
- Das, 2020
- Sethi et al, 2021
- Way Tan et al, 2021
Furthermore, the manuscript has been modified as reported:
“Several studies focus the attention on the influence of the Covid-19 on the EDs and ICUs (Bovim et al., 2022; Garcia-Vicuna et al., 2020; Le Lay et al., 2020; Melman et al., 2021) while less emphasis is given to outpatient services (Das, 2020; Sethi et al., 2021; Way Tan et al., 2021). This is a consequence of the non-urgent services, being locked down for extended periods, while the attention is placed on EDs and ICUs, where the most immediate emergencies lie. For assuring a safe restart of the entire healthcare system, the reopening of outpatient clinics respecting the Covid-19 anti-contagion guidelines allows the reduction of the pressure on the hospitals.”
C: There are colours in picture 1, no explanation anywhere.
The colours in picture 1 have no meaningful function, they are just used to distinguish the different macro-activities of the process flow.
C: There is no literature analysis of similar studies.
Within the limitations of the bibliographic research carried out by the authors, just three papers focus the attention on the simulation and/or performance analysis of the outpatient clinics in Covid-19 setting. The three references are reported below:
- Das, 2020, comparably analysing the impact of Covid-19 on process performances in endoscopic clinics.
- Sethi et al, 2021, dedicated to the enhancement of patient safety during Covid-19 in ophthalmology clinics.
- Tan et al, 2021, dealing with the optimization of the patient flow within paediatric clinics.
In literature, there are several research papers about the application of discrete-event simulation tools in clinics without the consideration of the pandemic factor, which represents the aspect of novelty of the manuscript.
C: Correctly designed model, is there a clinic to compare it with?
During the development of the article the outpatient clinic under examination (GSD Smart Clinic) was not operative. Given the recent opening of the facility, future works will surely involve the collection of real data to corroborate the current study and the execution of further improvement analysis.
C: The combination of results and discussion is a bad idea. It is worth discussing the results and then discussing the discussion.
Results and discussion have been divided into two separate sections (4.1 Results and 4.2 Discussion).
C: No reference to the limitations of the study noted by the authors.
The limitations of the study have been added in Section 5 as follows:
“To conclude, the developed simulation model gives a general overview of the MRI diagnostic examination process, enabling to collect data useful for the KPI evaluation. Further studies should extend the analysis on the adequacy of the models with the in-tended applications from qualitative means to quantitative means. With the opening of the GSD SC, it will be possible to conduct a data validation, bypassing the main limi-tation of the model. Furthermore, the presence of other healthcare services within the facility might affect the described process. A further model implemented by logics and interactions related to the other departments and processes allows to obtain a more representative model.”
C: Distressingly few references to literature.
New references are inserted in the introduction as follows:
- Sethi, K., Levine, E.S., Roh, S., Marx, J.L., Ramsey, D.J., 2021. Modeling the impact of COVID-19 on Retina Clinic Performance. BMC Ophthalmol. 21. https://doi.org/10.1186/S12886-021-01955-X
- Way TAN, K., Keow GOH, B., Gunawan, A., Way, K., Keow, B., Way Tan, K., Keow Goh, B., 2021. Redesigning patient flow in paediatric eye clinic for pandemic using simulation.
- Ferrin, D.M., Miller, M.J., McBroom, D.L., 2007. Maximizing hospital finanacial impact and emergency department throughput with simulation, in: Proceedings - Winter Simulation Conference. pp. 1566–1573. https://doi.org/10.1109/WSC.2007.4419774
- Miller, M.J., Ferrin, D.M., Messer, M.G., 2004. Fixing the emergency department: A transformational journey with EDSIM. Proc. - Winter Simul. Conf. 2, 1988–1993. https://doi.org/10.1109/WSC.2004.1371560
- Miller, M.J., Ferrin, D.M., Szymanski, J.M., 2003. Simulating six sigma improvement ideas for a hospital emergency department. Winter Simul. Conf. Proc. 2, 1926–1929. https://doi.org/10.1109/WSC.2003.1261655
- Pidd, M., 2003. Tools for Thinking, Modelling in Management Thinking. Wiley.
- Günal, M.M., Pidd, M., 2010. Discrete event simulation for performance modelling in health care: A review of the literature. J. Simul. 4, 42–51. https://doi.org/10.1057/jos.2009.25
Round 2
Reviewer 1 Report
The authors have made the suggested corrections to the manuscript.
Author Response
The authors would like to thank the reviewer for the suggested implementations and the positive evaluation.
Reviewer 2 Report
1) if the colors are not important, please remove them.
2) if there is no swab reference because the clinic was not working, maybe you should wait with the publication until it starts working.
3) literature still disturbingly poor
Author Response
Response to Reviewer 2
C: if the colours are not important, please remove them.
Colours have been removed from the first two figures.
C: if there is no swab reference because the clinic was not working, maybe you should wait with the publication until it starts working.
After the initial interruption of the outpatient activities, the access to medical services did not require any tests for the SARS-COV-2 gene, e.g. swab. To date, no anti-covid-19 swab is required for MRI outpatient entrance. Therefore, in compliance with the past and the present safety guidelines (both those of the clinic under consideration and those defined at national level), no swab reference was mentioned in the manuscript. Furthermore, this kind of clinic does not provide the swab service.
C: literature still disturbingly poor
Additional literature references are added:
- Garcia-Vicuña, D.; Esparza, L.; Mallor, F. Hospital preparedness during epidemics using simulation: the case of COVID-19. Cent. Eur. J. Oper. Res. 2022, 30, 213–249, doi:10.1007/S10100-021-00779-W.
- Lu, Y.; Guan, Y.; Zhong, X.; Fishe, J.N.; Hogan, T. Hospital Beds Planning and Admission Control Policies for COVID-19 Pandemic: A Hybrid Computer Simulation Approach. IEEE Int. Conf. Autom. Sci. Eng. 2021, 2021-August, 956–961, doi:10.1109/CASE49439.2021.9551589.
- Shahverdi, B.; Miller-Hooks, E.; Tariverdi, M.; Ghayoomi, H.; Prentiss, D.; Kirsch, T.D. Models for Assessing Strategies for Improving Hospital Capacity for Handling Patients during a Pandemic. Disaster Med. Public Health Prep. 2022, doi:10.1017/DMP.2022.12.
However, the impact of the anti-COVID-19 guidelines on outpatient clinic performances is much less investigated compared to hospital emergency department and intensive care unit. Hence, the available literature references were already mentioned in the manuscript.
